# Tracking Sources and Fate of Groundwater Nitrate in Kisumu City and Kano Plains, Kenya

**Benjamin Nyilitya [1,2,*], Stephen Mureithi [2] and Pascal Boeckx [1]** 

[1] Isotope Bioscience Laboratory-ISOFYS, Department of Green Chemistry and Technology, Faculty of Bioscience Engineering, Ghent University, Coupure Links 653, 9000 Gent, Belgium; pascal.boeckx@ugent.be

[2] Department of Land Resource Management and Agricultural Technology, University of Nairobi, P.O. Box 29053-00625 Nairobi, Kenya; stemureithi@uonbi.ac.ke

[*] Correspondence: Benjamin.nyilitya@ugent.be or kyalob73@yahoo.com; Tel.: +32-9-264-60-01

**Abstract:** Groundwater nitrate ($NO_3^-$) pollution sources and in situ attenuation were investigated in Kisumu city and Kano plains. Samples from 62 groundwater wells consisting of shallow wells (hand dug, depth <10 m) and boreholes (machine drilled, depth >15 m) were obtained during wet (May–July 2017) and dry (February 2018) seasons and analyzed for physicochemical and isotopic ($\delta^{15}N$-$NO_3^-$, $\delta^{18}O$-$NO_3^-$, and $\delta^{11}B$) parameters. Groundwater $NO_3^-$ concentrations ranged from <0.04 to 90.6 mg $L^{-1}$. Boreholes in Ahero town showed significantly higher $NO_3^-$ (20.0–70.0 mg $L^{-1}$) than boreholes in the Kano plains (<10.0 mg $L^{-1}$). Shallow wells in Kisumu gave significantly higher $NO_3^-$ (11.4–90.6 mg $L^{-1}$) than those in the Kano plains (<10.0 mg $L^{-1}$). About 63% of the boreholes and 75% of the shallow wells exceeded the drinking water WHO threshold for $NO_3^-$ and $NO_2^-$ (nitrite) during the study period. Mean $\delta^{15}N$-$NO_3^-$ values of 14.8‰ ± 7.0‰ and 20.7‰ ± 11.1‰, and $\delta^{18}O$-$NO_3^-$ values of 10.2‰ ± 5.2‰ and 13.2‰ ± 6.0‰ in wet and dry seasons, respectively, indicated manure and/or sewage as main sources of groundwater $NO_3^-$. However, a concurrent enrichment of $\delta^{15}N$ and $\delta^{18}O$ was observed, especially in the dry season, with a corresponding $NO_3^-$ decrease, indicating in situ denitrification. In addition, partial nitrification of mostly sewage derived $NH_4^+$ appeared to be responsible for increased $NO_2^-$ concentrations observed in the dry season. Specifically, targeted $\delta^{11}B$ data indicated that sewage was the main source of groundwater $NO_3^-$ pollution in shallow wells within Kisumu informal settlements, boreholes in Ahero, and public institutions in populated neighborhoods of Kano; while manure was the main source of $NO_3^-$ in boreholes and shallow wells in the Kano and planned estates around Kisumu. Waste-water sanitation systems in the region should be urgently improved to avoid further deterioration of groundwater sources.

**Keywords:** Kenya; Kisumu; groundwater; nitrate; hydrochemistry; denitrification

## 1. Introduction

Africa is quickly urbanizing with cities in Sub-Saharan Africa (SSA), reported to have grown at an average rate of 4.0% per annum over a period of 20 years, compared to the global average annual urban population growth rate of between 1.44% and 1.84% [1]. This rapid urbanization has put pressure on water service provision and sanitation infrastructure in SSA cities. Their sanitation infrastructure largely remains unchanged due to low capital investments, estimated at about 20% of the GDP. As utilities responsible for piped water and sanitation services struggle to meet the demand, community, and private-owned groundwater wells becomes the alternative option, not only to the rural population but also to urban dwellers. At the same time, nitrate ($NO_3^-$) is emerging as the most widespread groundwater contaminant associated with anthropogenic activities. The World Health Organization

(WHO) has put the maximum allowable concentration at 50 mg L$^{-1}$ nitrate. This is because high nitrate concentrations in water are both a health and environmental hazard promoting eutrophication, pose high risks to methemoglobinemia (blue baby syndrome) in infants and colorectal cancer [2,3].

In Kisumu city, which is located at the shores of Lake Victoria, groundwater use for domestic, industrial, and agricultural purposes is high. This is driven by rapid urbanization and industrialization due to a vibrant sugarcane production industry and a high population density ranging 827–4737 people per square kilometer, compared to Kenya's average population density of 82 people per square kilometer [4]. The supply of Lake Victoria water to the city is limited because of the low production capacity of the city's water service provider grappling with increased treatment costs caused by pollution and eutrophication of Lake Victoria. This has made the piped lake water unaffordable to the majority of the residents in the city and its surrounding areas, leading to an increased reliance on hand dug shallow wells or community water supply boreholes. Due to the minimal costs involved in their construction, shallow wells are widely used amongst urban residents in Kisumu and their distribution and significance has been well documented [5]. Despite the rapid expansion of the city, there has not been a corresponding investment in necessary waste management infrastructure, an occurrence that is characteristic of many other SSA cities [6]. The situation has been worsened by the proliferation of informal settlements (slums), which are not connected to any conventional sanitation system. The slum areas are characterized by the use of pit latrines, open defecation by both humans and animals, and landfills developing near groundwater resources. A similar sanitary situation is found in the rural Kano, although with a lower population density. In addition, the use of inorganic fertilizer and animal manure in the Kano area for sugarcane farming (Figure 1) may be a potential source of groundwater nitrate contamination.

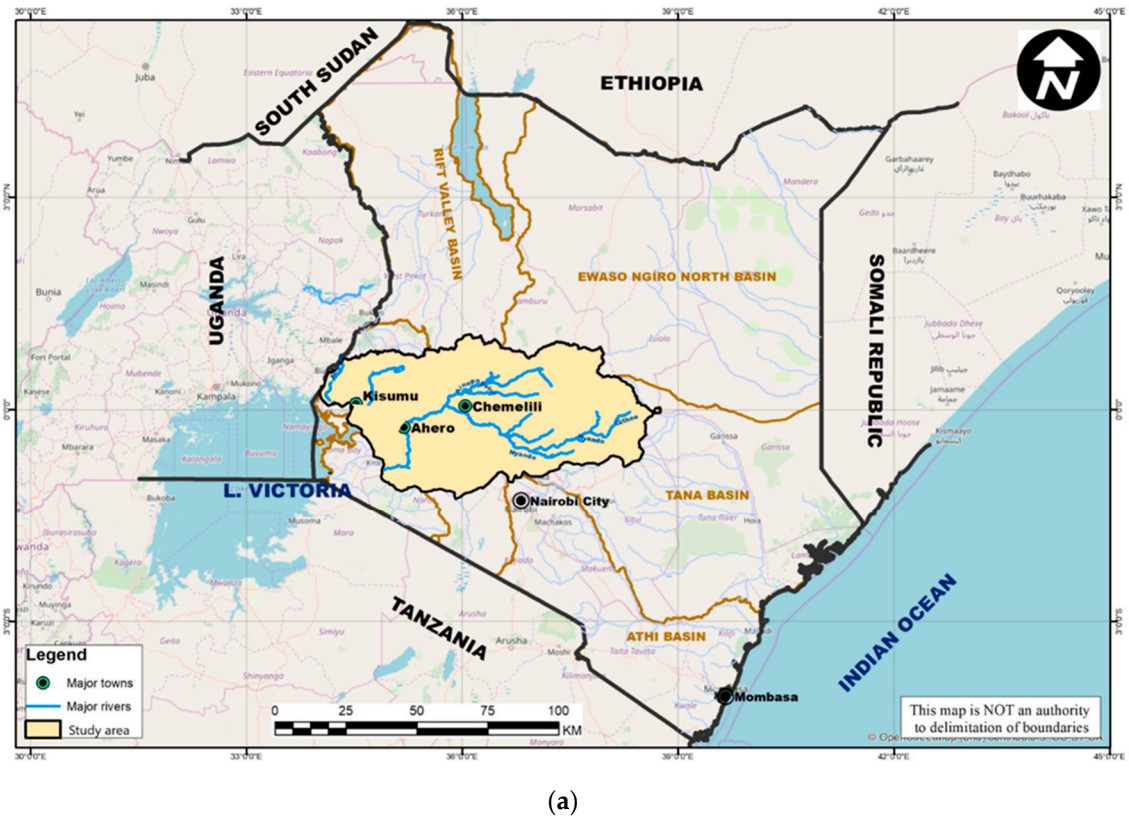

(**a**)

**Figure 1.** *Cont.*

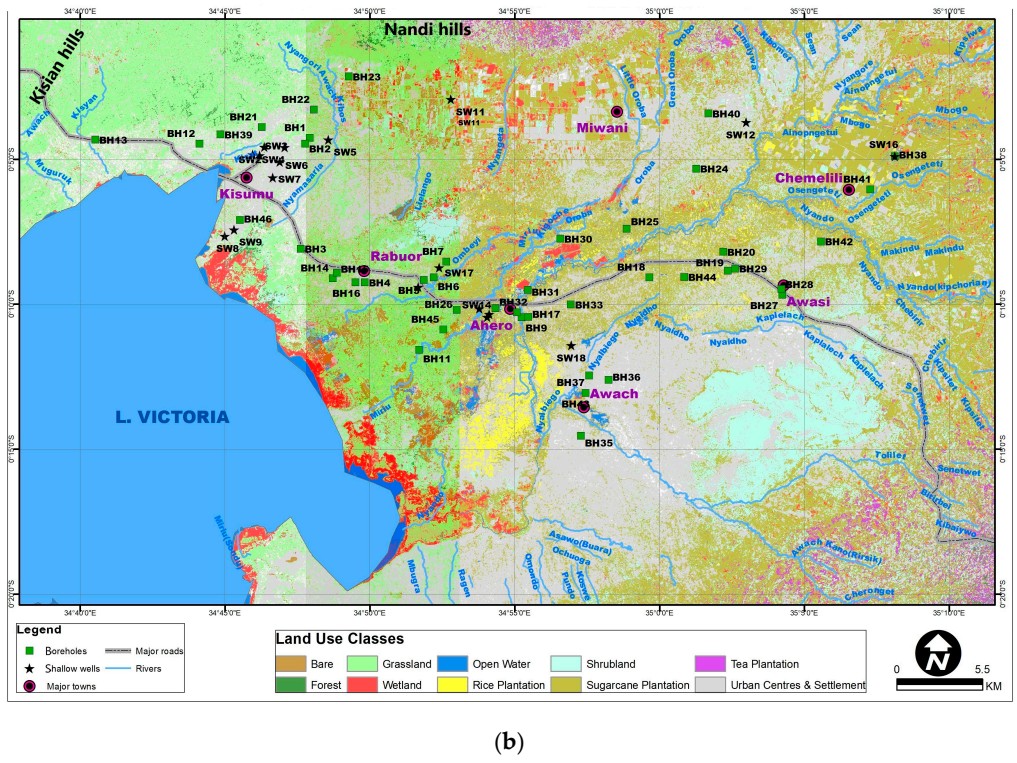

(**b**)

**Figure 1.** Study area map with an inset: location of the study area in the Lake Victoria basin, Kenya (**a**) and land use map indicating the spatial distribution of the groundwater sampling points represented by squares and stars for boreholes (BH) and shallow wells (SW), respectively (**b**).

Despite the high risks to groundwater nitrate contamination, the available information on groundwater nitrate in this part of the Lake Victoria basin is scarce [5,7]. Therefore, data on spatial nitrate distribution in the city and its surroundings is lacking and no study has attempted to identify the potential sources of groundwater nitrate or its fate. Coupling of hydrochemistry and nitrate isotopes ($\delta^{15}$N- and $\delta^{18}$O-$NO_3^-$) has demonstrated its usefulness in identifying potential $NO_3^-$ sources in groundwater and surface water [8]. In addition, the use of $\delta^{15}$N- and $\delta^{18}$O-$NO_3^-$ has proven to be a powerful method to pinpoint occurrence of $NO_3^-$ transformation processes such as nitrification and denitrification, influencing the $NO_3^-$ concentration in groundwater [9,10]. However, mixing of different $NO_3^-$ sources can lead to modification of the isotopic composition of dissolved $NO_3^-$ and at the same time, $\delta^{15}$N- and $\delta^{18}$O-$NO_3^-$ cannot clearly distinguish manure from sewage nitrate sources, since both sources have overlapping isotopic signatures [2]. Similarly, N transformation via nitrification and denitrification results into isotopic fractionations, which alter the isotopic signatures ($\delta^{15}$N and $\delta^{18}$O) of $NO_3^-$. These confounding factors complicate discrimination between multiple nitrate sources based on their isotopic composition. However, boron is a ubiquitous tracer in nature and is usually found in groundwater as a minor constituent. The large range of boron isotope ($\delta^{11}$B) ratios, observed in nature, enables clear contrasts to be made between boron sources in groundwater. Furthermore, boron is not affected by N transformation processes and researchers have demonstrated the added value of combining nitrate and boron isotope ($\delta^{15}$N, $\delta^{18}$O-$NO_3^-$, and $\delta^{11}$B) ratios in $NO_3^-$ source discrimination [11–14].

Together with providing baseline spatiotemporal water quality and isotopic data of the investigated area, this study seeks (1) to establish the potential sources of $NO_3^-$ input into groundwater and (2) to assess in situ attenuation controlling groundwater $NO_3^-$ concentration in the region. This information is paramount for policy development regarding groundwater use, sanitation, urban waste management, and use of agricultural inputs in the region.

## 2. Materials and Methods

### 2.1. Study Area

Kisumu city and the surrounding Kano plains are geographically located in a central plain surrounded by higher areas in the north-west (Kisian hill) and north-east (Nandi hills), bordering the Winam gulf of Lake Victoria (Figure 1). The city lies between latitude 00°02′ N–00°11′ S and longitude 34°35′–34°55′ E and covers approximately 417 km$^2$, and 35.5% is part of the Lake Victoria water body. The city has experienced a high population growth with a reported population of 567,963 according to the recent 2019 census [4]. It is Kenya's third largest city and the second most important city in the lake basin after Kampala [15]. The area receives a mean annual rainfall of 1245 mm occurring in two seasons, long and short rains from March–July and September–November, respectively. In addition, the mean annual minimum and maximum temperatures in the area are 17.3 and 28.9 °C, respectively [16]. The geology and hydrogeology of the study area has been described by Olago [17]. Kisumu city consists of fractured basalt overlain by pyroclastic deposits, which results into perched/unconfined aquifers with a localized recharge [5]. The unconfined aquifer comprises the shallow groundwater in the area, which occurs at depths ranging between 5 and 25 m [7]. The fractures act as pathways for groundwater flow and recharge, posing a great risk to groundwater pollution. Deeper aquifers (25–60 m) in the area occur in faulted and fractured hard rocks and sediments along the gulf, while thicker and well developed aquifers occur in the Kano area at depths greater than 150 m [17]. A piezometric map of the study area by Oiro [18] shows groundwater flows from the high altitude areas towards Lake Victoria as discharge point. This is in agreement with Olago [17], who reported that the groundwater flow direction in the Kisumu regional aquifer is from the highland recharge areas towards the central part of the Kano plains and the lake.

Land use in the study area includes the urban and peri-urban zone, which is mainly concentrated in the Kisumu municipality, an area covering 297 km$^2$ [16]. The other urban centers are Ahero and Awasi, which are situated in the Kano plains and lie along the busy Kisumu-Nairobi highway (Figure 1b). Major industries include cotton mills, breweries, cement factory, and several sugar milling and agro-chemical production industries. Urban agriculture and livestock keeping is common in the area and has been documented [16]. Sugarcane farming is the main agricultural activity in the Kano area (Figure 1), and is practiced under both small-scale mixed farming (with food crops: maize, sorghum, and finger millet) and large-scale commercial farming. Irrigated rice farming is practiced in the Ahero area next to the river Nyando.

### 2.2. Water Sampling and Analysis

In order to understand groundwater nitrate concentration, distribution, potential sources, and fate, the study targeted both the shallow unconfined aquifer where shallow wells (SW) tap from, and the deeper unconfined and confined aquifers where the boreholes (BH) are sank at depths above 25 m. The groundwater points were mapped and selected in May 2017 in consultation with the Kisumu-based water resources authority (WRA) regional office. The aim was to identify production BHs and SWs spatially distributed and representative of the major land use patterns in the area (Figure 1). The final list of sampling points included BHs and SWs in informal settlements (Ombuga, Nyalenda, and Manyatta) and formal settlements (Migotsi and Kibos) of Kisumu city where groundwater use is high. In the Kano area, BHs and SWs were selected from public institutions (e.g., schools), community donor funded projects, and private owned wells. A total of 62 groundwater production points were sampled during the wet season (May–July 2017) and in the dry season (February 2018).

During sampling, water samples were taken from production BHs and SWs. In case the well was not pumping prior to the sampling exercise, the well was purged to ensure the representativeness of a sample. Samples were first pre-filtered onsite using 11 μm filters (Whatman, GE Healthcare Life Sciences, Chicago, IL, USA) and then filled in a 200 mL PVC bottle after pre-rinsing with the sample water. Sample water was stored in an insulated cooler box containing ice cubes for transportation

to the laboratory for physicochemical and isotopic analysis. Duplicate samples for cation analysis were taken in 100 mL HDPE bottles, pre-filtered, and acidified to pH = 2 using diluted hydrochloric acid. In situ measurements included temperature (T), electrical conductivity (EC), pH, and dissolved oxygen (DO). These in situ parameters (T, EC, pH, and DO) were measured in every sampling station using a multi-parameter sensor (Multi3430, WTW, Germany). In the laboratory, all samples were filtered through 0.45 μm membrane filters and stored frozen (−17 °C) awaiting analysis. Laboratory determination of $Na^+$, $K^+$, $Ca^{2+}$, $Mg^{2+}$, $NH_4^+$, $NO_3^-$, $NO_2^-$, $Cl^-$, and $SO_4^{2-}$ concentrations was carried out using an ion chromatograph (930 Compact IC Flex, Metrohm, Switzerland).

The $\delta^{15}$N- and $\delta^{18}$O-$NO_3^-$ values were determined by the "Bacterial denitrification method" [19–21], which allows simultaneous determination of $\delta^{15}$N and $\delta^{18}$O in $N_2O$ produced from the conversion of $NO_3^-$ by denitrifying bacteria, which naturally lack $N_2O$-reductase activity. The $\delta^{15}$N and $\delta^{18}$O analysis of the produced $N_2O$ was carried out using a trace gas preparation unit (ANCA TGII, SerCon, UK), coupled to an isotope ratio mass spectrometer (IRMS; 20–20, SerCon, UK). The $N_2O$ sample was flushed out of the sample vial using a double-hole needle on an auto-sampler. Water was removed using a combination of nafion dryer and $MgClO_4$ scrubber. By cryogenic trapping and focusing, the $N_2O$ was compressed onto a capillary column (CP-Poraplot Q 25 m, 0.32 mm id, 10 μm df, Varian, US) at 35 °C and subsequently analyzed by IRMS. The subsequent stable isotope data were expressed as delta ($\delta$) units in per mil (‰) notation relative to the respective international standards:

$$-\delta_{sample}\ (‰) = \left[\frac{R_{sample}}{R_{standard}} - 1\right] \times 1000 \tag{1}$$

where $R_{sample}$ and $R_{standard}$ are the $^{15}$N/$^{14}$N or $^{18}$O/$^{16}$O ratio of the sample and the standard for $\delta^{15}$N and $\delta^{18}$O, respectively. $\delta^{15}$N values are reported relative to $N_2$ in atmospheric air (AIR) and $\delta^{18}$O are reported relative to Vienna Standard Mean Ocean Water (VSMOW). Three internationally recognized reference standards, USGS32 (180.0‰ ± 1.0‰ for $\delta^{15}$N, 25.7‰ ± 0.4‰ for $\delta^{18}$O), USGS34 (−1.8‰ ± 0.2‰ for $\delta^{15}$N, −27.8‰ ± 0.4‰ for $\delta^{18}$O), and USGS35 (2.7‰ ± 0.2‰ for $\delta^{15}$N, 56.8‰ ± 0.3‰ for $\delta^{18}$O), were used to normalize the raw $\delta^{15}$N- and $\delta^{18}$O-$NO_3^-$ values (based on a $N_2O$ reference gas tank) to the AIR and VSMOW scale. USGS32 and USGS34 were used for normalization of the $\delta^{15}$N value and USGS34 and USGS35 for the $\delta^{18}$O. The amount of $NO_3^-$ in samples and references were matched (i.e., 20 nmol), which corrects for nonlinearity of the IRMS and blanks associated with the procedure. An in-house $KNO_3$ laboratory standard (9.9‰ for $\delta^{15}$N, 24.3‰ for $\delta^{18}$O) was analyzed together with the samples for quality control. Measurement batches were only accepted if measured $\delta^{15}$N and $\delta^{18}$O values of the laboratory standard were within 0.4‰ and 0.5‰ of our accepted values, respectively. If standard deviation on replicate samples was higher than 0.3 and 0.4 for $\delta^{15}$N and $\delta^{18}$O, respectively, the sample was reanalyzed. This method is well explained in [20,21].

The water analysis technique for B and $\delta^{11}$B is well covered in [22]. Samples underwent a two-step chemical purification using Amberlite IRA-743-selective resin (method adapted from Gaillardet and Allegre [23]). First, the sample (pH = 7) was loaded on a Teflon PFA®column filled with 1 mL resin, previously cleaned with ultrapure water and 2 N ultrapure NaOH. After cleaning the resin again with water and NaOH, the purified B was collected with 15 mL of sub-boiled HCl 2 N. After neutralization of the HCl with Superpur $NH_4OH$ (20%), the purified B was loaded again on a small 100 mL resin Teflon PFA®column. B was collected with 2 mL of HCl 2N. An aliquot corresponding to 2 mg of B was then evaporated below 70 °C with mannitol ($C_6H_8(OH)_6$) in order to avoid B loss during evaporation [24]. The dry sample was loaded onto a tantalum (Ta) single filament with graphite (C), mannitol, and cesium (Cs). $\delta^{11}$B values were then determined by measuring the $Cs_2BO_2^+$ ion [25,26] by a thermal ionization mass spectrometer. The analysis was run in the dynamic mode by switching between masses 308 and 309. Each analysis corresponded to 10 blocks of 10 ratios. Samples were always run twice. Total B blank was less than 10 ng corresponding to a maximum contribution of 0.2%, which is negligible. Seawater (IAEA-B1) was purified regularly in the same way, in order to check for possible chemical fractionation due to an uncompleted recovery of B, and to evaluate the accuracy

and reproducibility of the overall procedure (e.g., [27]). Reproducibility was obtained by repeated measurements of the NBS951, and the accuracy was controlled with the analysis of the IAEA-B1 seawater standard ($\delta^{11}B = 38.6‰ \pm 1.7‰$). Similar to N and O, B isotope ratios were expressed in delta ($\delta$) units and a per mil (‰) notation relative to an international standard, NBS951.

Statistical analysis for determining spatial-temporal differences in physico-chemical and isotopic parameters was performed using ANOVA with Tukey HSD tests, and $p < 0.05$ significance level was set to test significance between the parameter sets.

## 3. Results and Discussion

### 3.1. Hydrochemistry

Table 1 presents a statistical summary of 16 hydro-chemical and three isotopic parameters for boreholes and shallow wells sampled during the wet and dry seasons. Water pH ranged from 6.1 to 10.1 and from 6.2 to 8.6 in boreholes and shallow wells, respectively. Two borehole samples (BH 27 and 28) exceeded the World Health Organization (WHO) pH limit (>8.5) during both wet and dry seasons while shallow wells within Kisumu recorded a slightly acidic pH (<6.5). High pH observed in BH 27 and 28, located in the Awasi area (Figure 1b), is characteristic of cation exchange reactions where $Ca^{2+}$ and/or $Mg^{2+}$ in solution is exchanged with $Na^+$ on clay minerals [28]. Replacement of $Ca^{2+}$ by $Na^+$ in solution results into a pH change, which is sanctioned by change in the equilibrium of the reaction:

$$CaCO_3 + CO_2(g) + H_2O = Ca^{2+} + 2HCO_3^- \tag{2}$$

It shifts the equilibrium further to the right and increases the bicarbonate content and pH. This is supported by the significantly low $Ca^{2+}$ ($p < 0.0001$) and $Mg^{2+}$ ($p = 0.04$) values ranging from 1.1 to 1.7 mg $L^{-1}$ and from 0.1 to 0.2 mg $L^{-1}$, respectively obtained in the two boreholes during the study period (Supplementary Tables S1 and S2). Consequently, high pH and low calcium levels favor the solubility of fluoride, and this accounts for the high fluoride levels (9.0 and 11.0 mg $L^{-1}$, respectively; Supplementary Table S1) observed in the two Awasi boreholes. Generally, groundwater in the area had high fluoride levels as shown in Table 1 where mean values exceeded the WHO limit (1.5 mg $L^{-1}$) for drinking water. This is a health risk since excessive intake of fluoride is known to cause dental/skeletal fluorosis, cancer, low hemoglobin levels, osteoporosis, reduced immunity, and thyroid disorders [29].

Water temperature ranged from 25.3 to 37.6 °C and from 24.5 to 28.5 °C in boreholes and shallow wells, respectively, and did not significantly differ by season. The EC values ranged from 295 to 2562 µS cm$^{-1}$ in boreholes and from 248 to 1427 µS cm$^{-1}$ in shallow wells portraying an increase in mineralization with well depth. On the other hand, dissolved oxygen (DO) values ranged from 1.4 to 6.8 mg $O_2$ $L^{-1}$ in boreholes and from 1.2 to 9.8 mg $O_2$ $L^{-1}$ in shallow wells. However, the wet season showed lower DO values than the dry season in both boreholes and shallow wells. This implies increased oxygen consumption in groundwater most likely by dissolved organic carbon originating from contaminated surface water during the wet season. Nitrate showed a wide variation ranging from <0.04 to 70.0 mg $L^{-1}$ and from <0.04 to 90.6 mg $L^{-1}$ in boreholes and shallow wells, respectively (Table 1). Seasonally, $NO_3^-$ concentrations in shallow wells were significantly higher in the wet season compared to the dry season with about 70% of shallow wells located in Kisumu city giving values above the WHO recommended limit (50 mg $L^{-1}$). Unlike $NO_3^-$, $NO_2^-$ concentration was significantly higher in the dry season than in the wet season for both boreholes and shallow wells with 60% of the samples exceeding the WHO limit (0.2 mg $L^{-1}$) during the dry season. The biogeochemical processes governing the $NO_3^-$ and $NO_2^-$ variations are discussed further in Section 3.3. In general, about 63% of the boreholes and 75% of the shallow wells exceeded the WHO recommended limit values for $NO_3^-$ and $NO_2^-$ during the study period. Both boreholes and shallow wells showed similar concentrations of the major anions: $Cl^-$, $SO_4^{2-}$, and $HCO_3^-$ and the cations: $Na^+$, $K^+$, $Ca^{2+}$, $Mg^{2+}$, and $NH_4^+$ during wet and dry seasons with values falling within the WHO limits for drinking water.

**Table 1.** Statistical summary of hydro-chemical and isotopic parameters for boreholes ($n = 44$) and shallow wells ($n = 18$) sampled during wet and dry seasons. WHO limit stands for World Health Organization standard limits for drinking water; DO stands for dissolved oxygen; $p$-value represents the ANOVA (at $p < 0.05$ levels) output for testing differences of parameters between wet and dry seasons.

| | Boreholes | | | | | | p Value | WHO Limit | Number Exceeded WHO | Shallow Wells | | | | | | p Value | WHO Limit | Number Exceeded WHO |
|---|---|---|---|---|---|---|---|---|---|---|---|---|---|---|---|---|---|---|
| | Wet | | | Dry | | | | | | Wet | | | Dry | | | | | |
| | Min | Mean | Max | Min | Mean | Max | | | | Min | Mean | Max | Min | Mean | Max | | | |
| **Physico-Chemical Parameters** | | | | | | | | | | | | | | | | | | |
| pH (-) | 7 | $7.6 \pm 0.7$ | 10.1 | 6.1 | $7.4 \pm 0.6$ | 9.6 | 0.23 | 6.5–8.5 | 4 | 6.3 | $7.1 \pm 0.7$ | 8.6 | 6.2 | $7.0 \pm 0.6$ | 8.2 | 0.70 | 6.5–8.5 | 10 |
| Temp (°C) | 25.3 | $28.0 \pm 2.2$ | 37.6 | 25.7 | $28.1 \pm 1.9$ | 36 | 0.84 | - | 0 | 25.1 | $26.4 \pm 0.9$ | 28.2 | 24.5 | $26.2 \pm 1.0$ | 28.5 | 0.43 | - | 0 |
| EC ($\mu$S cm$^{-1}$) | 295 | $1091 \pm 390$ | 2520 | 400 | $1052 \pm 383$ | 2562 | 0.71 | - | 0 | 248 | $821 \pm 282$ | 1420 | 290 | $785 \pm 3.02$ | 1427 | 0.74 | - | 0 |
| DO (mg O$_2$ L$^{-1}$) | 1.4 | $3.1 \pm 1.4$ | 6.6 | 2 | $3.9 \pm 1.2$ | 6.8 | 0.01 | - | 0 | 1.2 | $2.9 \pm 1.5$ | 6.2 | 1.6 | $4.1 \pm 2.0$ | 9.8 | 0.05 | - | 0 |
| Cl$^-$ (mg L$^{-1}$) | 1.6 | $22 \pm 21$ | 80.1 | 0.1 | $29.4 \pm 39.7$ | 156 | 0.29 | 250 | 0 | 1.7 | $32.0 \pm 22.8$ | 75.5 | 2.8 | $41.5 \pm 34.3$ | 103 | 0.33 | 250 | 0 |
| SO$_4^{2-}$ (mg L$^{-1}$) | 0.9 | $37 \pm 41$ | 212 | 0.4 | $49.1 \pm 67.7$ | 360 | 0.33 | 250 | 0 | 1.4 | $31 \pm 17$ | 56.3 | 9 | $37 \pm 17.5$ | 67.2 | 0.38 | 250 | 0 |
| NO$_3^-$ (mg L$^{-1}$) | <0.04 | $5.8 \pm 8.8$ | 43.7 | <0.04 | $6.7 \pm 12.5$ | 69.9 | 0.74 | 50 | 1 | <0.04 | $33.5 \pm 32.4$ | 90.6 | 0.04 | $10.9 \pm 13.0$ | 38.2 | 0.02 | 50 | 6 |
| NO$_2^-$ (mg L$^{-1}$) | <0.04 | $0.02 \pm 0.01$ | 0.06 | <0.04 | $1.4 \pm 0.9$ | 3.2 | <0.0001 | 0.2 | 27 | <0.04 | $0.04 \pm 0.04$ | 0.15 | <0.04 | $0.6 \pm 0.8$ | 2.2 | 0.006 | 0.2 | 6 |
| HCO$_3^-$ (mg L$^{-1}$) | 2.4 | $93 \pm 46$ | 167 | 20.2 | $98.5 \pm 36.4$ | 172 | 0.67 | 500 | 0 | 16.6 | $47 \pm 38$ | 109 | 4.9 | $57.2 \pm 43.8$ | 122 | 0.61 | 500 | 0 |
| Na$^+$ (mg L$^{-1}$) | 33.6 | $190 \pm 92$ | 452 | 29.8 | $186 \pm 98$ | 461 | 0.86 | 250 | 0 | 41 | $112 \pm 75$ | 311 | 31.1 | $98.3 \pm 63.0$ | 236 | 0.60 | 250 | 0 |
| K$^+$ (mg L$^{-1}$) | 4.9 | $23.6 \pm 15.5$ | 66.5 | 3.5 | $23.3 \pm 14.7$ | 67.9 | 0.90 | 250 | 0 | 1.3 | $22 \pm 15$ | 55 | 3.1 | $23.8 \pm 17.8$ | 62.1 | 0.75 | 250 | 0 |
| Ca$^{2+}$ (mg L$^{-1}$) | 1.11 | $23.9 \pm 16.6$ | 74.7 | 1.2 | $23.2 \pm 13.3$ | 60 | 0.85 | 75 | 0 | 2.5 | $26.6 \pm 14.2$ | 51.7 | 7.7 | $24.7 \pm 12.2$ | 42.4 | 0.70 | 75 | 0 |
| Mg$^{2+}$ (mg L$^{-1}$) | 0.1 | $6.1 \pm 7.1$ | 31.3 | 0.04 | $7.4 \pm 7.8$ | 30 | 0.46 | 50 | 0 | 0.4 | $8.0 \pm 6.8$ | 30 | 1.8 | $7.1 \pm 3.8$ | 12.9 | 0.68 | 50 | 0 |
| NH$_4^+$ (mg L$^{-1}$) | <0.01 | $0.03 \pm 0.02$ | 0.14 | <0.01 | $0.03 \pm 0.02$ | 0.09 | 0.84 | - | 0 | <0.01 | $0.7 \pm 2.8$ | 11.4 | <0.01 | $1.1 \pm 3.9$ | 14.1 | 0.74 | - | 0 |
| B ($\mu$g L$^{-1}$) | 16 | $24.5 \pm 7.5$ | 34 | | | | | - | 0 | 20 | $23 \pm 4.2$ | 26 | | | | | - | 0 |
| F$^-$ (mg L$^{-1}$) | 0.7 | $4.1 \pm 2.6$ | 10.9 | | | | | 1.5 | 31 | 0.3 | $1.6 \pm 2.0$ | 8.0 | | | | | 1.5 | 6 |
| **Isotopic Parameters** | | | | | | | | | | | | | | | | | | |
| $\delta^{15}$N (‰) | 4.1 | $12.6 \pm 6.2$ | 25.8 | 6.9 | $15.5 \pm 6.0$ | 32.2 | 0.06 | - | | 8 | $19.5 \pm 5.8$ | 28.9 | 12.4 | $33.6 \pm 11.5$ | 51.8 | 0.0004 | - | |
| $\delta^{18}$O (‰) | −2.4 | $9 \pm 5.4$ | 20.8 | −1.7 | $11.1 \pm 5.5$ | 24.1 | 0.13 | - | | 7.5 | $13.3 \pm 4.0$ | 19.8 | 12 | $18.0 \pm 4.8$ | 29.3 | 0.0138 | - | |
| $\delta^{11}$B (‰) | 23 | $30.3 \pm 6.3$ | 36 | | | | | - | | 16 | $25 \pm 12.7$ | 34 | | | | | - | |

In Figure 2, water facies are presented in a Piper diagram [30] for the boreholes and shallow wells. The boreholes and shallow wells were grouped into spatial categories, which also display similarities in hydro-chemical and isotopic parameters. These were shallow wells located in Kisumu city (SW-Kisumu city); shallow wells located in Kano plain (SW-Kano plain); boreholes located in Ahero town (BH-Ahero town); boreholes located in public institutions in highly populated neighborhoods (BH-Public); and boreholes located in the Kano plain (BH-Kano plain). It is clear that the groundwater positioning in the Piper diagram is mainly determined by hydrogeology and anthropogenic activities. A mineralization trend was observed (Figure 2, cation triangle) from the low mineralized boreholes (BH: 13, 23, and 41) located at the foot of the Nandi and Kisian hills (recharge areas) to highly mineralized boreholes located around Awasi town (BH: 27, and 28). While this, on the one hand, represents cation exchange of $Ca^{2+}$ by $Na^+$ in solution as discussed earlier, on the other hand, it is an indicator of the direction of groundwater recharge. According to Olago [17], groundwater recharges from the high altitude areas of the Nandi hills towards the center of the Kano plain (Awasi). These boreholes located near the recharge zones (Figure 1b) had a $Ca^{2+}$-$Mg^{2+}$-$HCO_3^-$ and $Na^+$-$Ca^{2+}$-$Mg^{2+}$-$HCO_3^-$ water type representing fresh and recharging groundwater [31]. The same water type was obtained in BH 42, which by being located next to R. Nyando (Figure 1b) revealed young/recharging water from the river [32]. However, the Awasi boreholes (BH: 27 and 28) with a high pH showed a dominant $Na^+$-$Cl^-$ water type by plotting on the far-right corner of the diamond. In addition, these boreholes had a low $NO_3^-$ content ($<1$ mg $L^{-1}$), but relatively high EC ($>1000$ µS $cm^{-1}$), characterizing saline water [28,30].

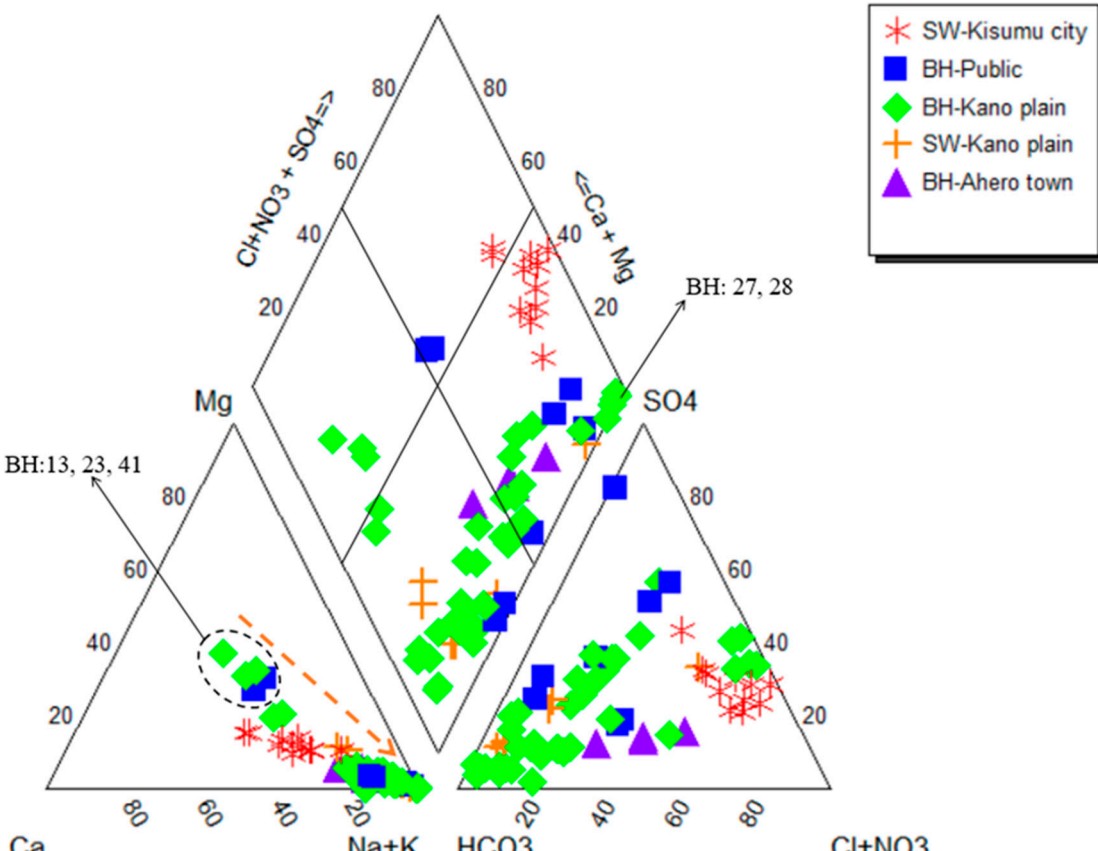

**Figure 2.** Piper diagram representation of major cations and anions (in % meqL$^{-1}$) for water characterization. Samples are categorized as, SW-Kisumu city: shallow wells located in Kisumu city; SW-Kano plain: shallow wells located in Kano plain; BH-Ahero town: boreholes located in Ahero town; BH-Public: boreholes in public institutions located in populated neighborhoods; and BH-Kano plain: boreholes located in the Kano plain. BH 13, 23, and 41 are located near Nandi/ Kisian hills; BH 27 and 28 are located in Awasi town; dotted brown arrow shows increasing groundwater mineralization.

The shallow wells in the diamond grouped into two categories: a group consisting of the SW-Kisumu city showing a $Na^+$-$K^+$-$Cl^-$-$NO_3^-$ or $Na^+$-$K^+$-$Cl^-$-$SO_4^{2-}$ water type, and a group made of the SW-Kano plain showing a $Na^+$-$K^+$-$HCO_3^-$ or $Na^+$-$HCO_3^-$ water type. The majority of the BH-Kano plain, just like the shallow wells, had the $Na^+$-$K^+$-$HCO_3^-$ and $Na^+$-$HCO_3^-$ water type. Previous studies conducted in the study area by Oiro [18] similarly report $Na^+$ and $HCO_3^-$ as the dominant cation and anion respectively in borehole water. The study also observed high $NO_3^-$ and $SO_4^-$ levels in shallow wells located in Kisumu. This suggests that weathering of $Na^+$-$K^+$ containing alumino-silicate minerals is the major process contributing to the dominance of $Na^+$ and $K^+$ in the study area. In addition, the dissolution of $CO_2$ and carbonate during precipitation and infiltration in the unsaturated zone impart the bicarbonate character of the groundwater [33]. However, the dominance of $NO_3^-$, $Cl^-$, and $SO_4^{2-}$ anions observed in SW-Kisumu city suggests the influence of contaminated surface water, just as alluded in the previous study [18]. This is in agreement with a similar study by Kamtucheng et al. [34] on surface and groundwater hydrogeochemistry around Lake Manoun, Cameroon. The study observed $Na^+$-$K^+$-$Cl^-$-$NO_3^-$ and $Ca^{2+}$-$Mg^{2+}$-$Cl^-$-$NO_3^-$ as the water types representing freshwater mixed with contaminated sources. The rest of the BH-Kano plain, BH-Public, and BH-Ahero displayed a $Na^+$-$K^+$-$Cl^-$-$SO_4^{2-}$ or $Na^+$-$K^+$-$Cl^-$-$NO_3^-$ water type.

### 3.2. Spatial Groundwater $NO_3^-$ Distribution and Its Controlling Factors

Figure 3 presents the spatial $NO_3^-$ concentration ranges (averaged over the wet and dry seasons) obtained in the boreholes and shallow wells. It was observed that the highest $NO_3^-$ concentration in shallow wells, above the WHO limit (50–91 mg $L^{-1}$), was mainly recorded in Kisumu city. These account for 70% of the shallow wells sampled in the city and are located in informal settlements (Ombuga, Nyalenda, and Manyatta) where locals depend on shallow groundwater because of low costs associated with their construction [5]. The rest of the shallow wells in the city had $NO_3^-$ concentrations ranging between 10 and 50 mg $L^{-1}$ and are situated in newly planned but also highly populated estates (Kibos and Migotsi). Kisumu is a densely populated city in Kenya with an average city population density of 2375 people per $km^2$, compared to Kenya's average density of 82 people per $km^2$ [4]. Due to the limited sewerage infrastructure network in Kisumu, untreated sewage discharges into urban streams is common. In addition, the majority of the urban population living in the informal settlements use pit latrines [5]. This can easily result in leaching of $NO_3^-$ into the unconfined shallow aquifer system of the city. The SW-Kano plain had lower $NO_3^-$ concentrations with most these wells recording values < 10 mg $L^{-1}$ except for SW 17, and 18 which gave > 10 mg $L^{-1}$ during the wet season. During the dry season, however, a general decrease of $NO_3^-$ concentrations was observed in shallow wells to values ranging between 0.04 and 38.2 mg $L^{-1}$ (Table 1).

On the other hand, $NO_3^-$ concentrations in boreholes varied spatially with BH-Ahero town samples recording significantly higher $NO_3^-$ concentrations (20.0–69.9 mg $L^{-1}$) in both seasons, than those located elsewhere in the study area. The relatively high $NO_3^-$ in Ahero, where one exceeded the WHO threshold should raise concern regarding $NO_3^-$ pollution management in the town because the three main water supply wells to the town were sampled. Two of them belong to the local water service provider and the third is run by a faith-based organization. Ahero is a highly populated town located along the busy Kisumu-Nairobi highway and totally lacking a conventional sewage management system. In addition, irrigation rice farming is the common agricultural activity around Ahero (See Figure 1b) and uses water from the river Nyando, implying that the use of inorganic fertilizers (i.e., $(NH_4)_2SO_4$) may also be contributing to the observed $NO_3^-$ levels. $NO_3^-$ concentrations ranging from 10 to 50 mg $L^{-1}$ in boreholes were measured in public schools and in a community water supply (BH: 7, 13, 18, 29, and 35).

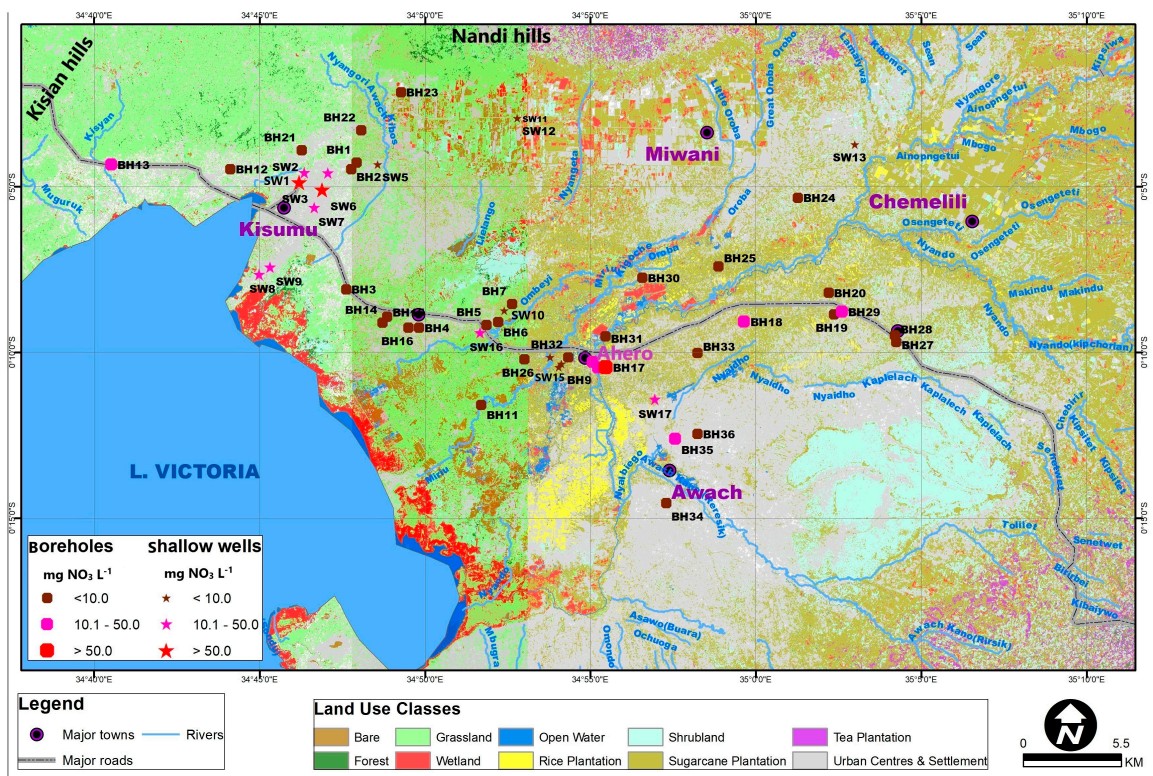

**Figure 3.** Spatial groundwater $NO_3^-$ concentration (average of wet and dry seasons) map of the study area represented in ranges by bullet sizes; squares for boreholes, stars for shallow wells.

The lowest $NO_3^-$ concentration in boreholes <10 mg $L^{-1}$ were recorded mainly in the less populated parts of the Kano plain, which spans from the outskirts of Kisumu, the sugarcane belt and the recharge areas near Nandi hills (Figure 3). Based on Figure 2, the low groundwater $NO_3^-$ range belongs to the $Ca^{2+}$-$Mg^{2+}$-$HCO_3^-$ and $Na^+$-$K^+$-$HCO_3^-$ water types, which revealed either a dilution from the low $NO_3^-$ recharging water or background $NO_3^-$ levels. However, as water changes to $Na^+$-$K^+$-$Cl^-$-$SO_4^{2-}$ or $Na^+$-$K^+$-$Cl^-$-$NO_3^-$ type, $NO_3^-$ concentration increased tremendously to levels > 20 mg $L^{-1}$. This points out to urbanization and human population pressure as the key drivers to groundwater $NO_3^-$ increase in the study area. The low $NO_3^-$ (<1 mg $L^{-1}$) reported in the Awasi boreholes (BH 27 and 28) was due to its confined and thick aquifer occurring at depths greater than 150 m [17].

### 3.3. Use of Multi Isotope and Hydro-Chemical Methods to Track Sources of Groundwater Nitrate Contamination and Removal

For the implementation of effective $NO_3^-$ pollution control strategies, spatial water quality data alone is not sufficient without identifying the potential $NO_3^-$ sources and associated biogeochemical processes controlling groundwater $NO_3^-$ concentration. In order to identify the sources of groundwater $NO_3^-$ input, $\delta^{15}N$- and $\delta^{18}O$-$NO_3^-$ values of boreholes and shallow wells, in addition to those of the potential $NO_3^-$ sources from the study area were determined. Isotopic values obtained for the local $NO_3^-$ sources: manure and sewage, soil N, precipitation, $NO_3^-$ fertilizer, and $NH_4^+$ in fertilizer and rainfall (Figure 4) were found to be within the literature ranges [2,19,35]. The $\delta^{15}N$-$NO_3^-$ in boreholes ranged from +4.1‰ to +25.8‰ and from +6.9‰ to +32.2‰ in wet and dry seasons, respectively. In shallow wells, $\delta^{15}N$-$NO_3^-$ ranged from +8.0‰ to +28.9‰ and from +12.4‰ to +51.8‰ during the wet and dry seasons, respectively (Table 1). $\delta^{18}O$-$NO_3^-$ in boreholes, on the other hand, ranged from −2.4‰ to +20.8‰ and from −1.7‰ to +24.1‰ during wet and dry seasons, respectively, while in

shallow wells, it ranged from +7.5‰ to +19.8‰ and from +12‰ to +29.3‰ during the wet and dry seasons, respectively.

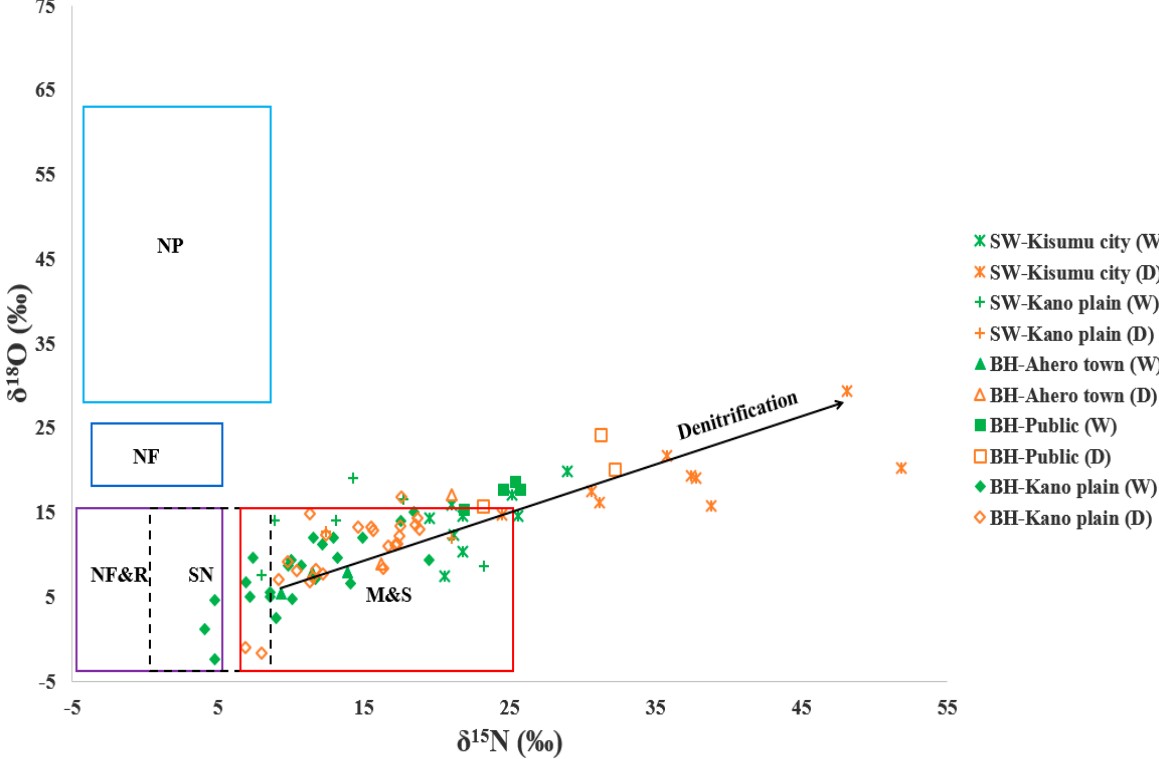

**Figure 4.** $\delta^{15}N$- vs. $\delta^{18}O$-$NO_3^-$ plot of groundwater samples categorized as, SW-Kisumu city: shallow wells located in Kisumu city; SW-Kano plain: shallow wells in Kano plain; BH-Ahero town: boreholes in Ahero town; BH-Public: boreholes in public institutions located in populated neighborhoods; and BH-Kano plain: boreholes in Kano plain, during wet (green symbols) and dry (brown symbols) seasons. Boxes represent $\delta^{15}N$- and $\delta^{18}O$-$NO_3^-$ ranges of local $NO_3^-$ sources: $NO_3^-$ in precipitation (NP), $NO_3^-$ fertilizer (NF), $NH_4^+$ in fertilizer and rainfall (NF&R), soil N (SN), and manure and sewage (M&S). Black arrow indicates a denitrification vector for the increasing $\delta^{15}N$ and $\delta^{18}O$.

A $\delta^{15}N$ vs. $\delta^{18}O$ plot of the groundwater samples during wet and dry seasons (Figure 4) clearly shows that the majority of the groundwater samples lie in the manure and/or sewage source range. However, a few exceptions of the BH-Kano plain samples (BH: 7, 18, and 20; see Figure 3) plotted in the mixed soil N and $NH_4^+$ in fertilizer and rainfall source range during the wet season. Isotopic enrichment with population density is evident from Figure 4 as shown by all the SW-Kisumu city, BH-Public and BH-Ahero town samples. These are the high population density areas (827–4737 people per square kilometer), which gave significantly higher $\delta^{15}N$ values ($p < 0.0001$: wet and dry seasons) compared to the Kano plain (BH-Kano plain and SW-Kano plain), an area with a lower population density ranging 234–362 people per square kilometer [4]. In addition, the SW-Kisumu city and BH-Ahero town with enriched $\delta^{15}N$ also recorded relatively high $NO_3^-$ concentrations above 20 mg L$^{-1}$ (Figures 3 and 4). This is an indication that sewage, characteristically enriched in $\delta^{15}N$ [19], is likely the driving force to groundwater $NO_3^-$ contamination in the major urban areas. A gradual enrichment in $\delta^{15}N$- and $\delta^{18}O$-$NO_3^-$ was observed in both boreholes and shallow wells moving from the wet to dry season (Figure 4). The isotopic enrichment observed in the dry season was accompanied by a $NO_3^-$ decrease in all of the SW-Kisumu city samples and some of the BH-Kano plain (BH: 1, 7, 18, 20, and 29; Figure 5). This indicates that in situ $NO_3^-$ attenuation via denitrification is taking place in the groundwater points during the dry season. Kinetic isotope effects during denitrification preferably convert lighter isotopes ($^{14}N$ and $^{16}O$) to $N_2$ and $N_2O$, causing an enrichment of the heavy isotopes

($^{15}$N and $^{18}$O) in the residual $NO_3^-$ [36,37]. It is also clear that denitrification was more pronounced in SW-Kisumu city, with high $NO_3^-$ concentrations in the wet season. Based on Figures 4 and 5, it appears that denitrification may also be responsible for the low $NO_3^-$ concentration (but high $\delta^{15}N$ and $\delta^{18}O$) values recorded in the BH-Public samples during both wet and dry seasons. In addition, a general linear relationship indicating an enrichment of $\delta^{15}N$ relative to $\delta^{18}O$ by a factor of between 1.3:1 and 2.1:1 supports evidence for in situ denitrification [9]. Figure 4 shows a linear relationship between $\delta^{15}N$ and $\delta^{18}O$ ($\delta^{18}O = 0.5\ \delta^{15}N + 2.9$, $R^2 = 0.7$) for all groundwater samples. The slope revealed an enrichment of $\delta^{15}N$ relative to $\delta^{18}O$ by a factor of 2:1, which is characteristic of denitrification [9,13,38]. Previous work done in the region by Nyilitya et al. [32] and involving one of the BH-Public wells (BH11, Figure 3), obtained $\delta^{15}N$ and $\delta^{18}O$ values of 18‰ ± 1.2‰ and 20‰ ± 0.2‰ respectively for the well, and a corresponding $NO_3^-$ concentration of 0.6 mg L$^{-1}$. In comparison to Nyando river and boreholes located in the headwater catchments, the BH-Public well had low $NO_3^-$ concentration and showed a linear isotopic enrichment, which the authors attributed to denitrification.

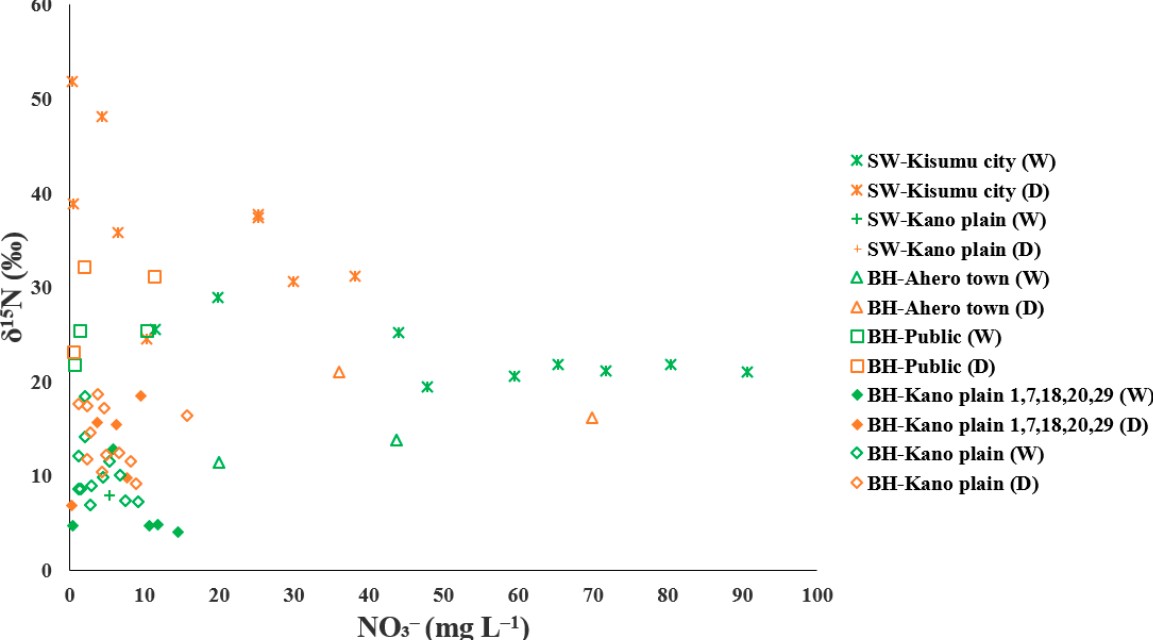

**Figure 5.** $NO_3^-$ concentrations vs. $\delta^{15}N$-$NO_3^-$ of groundwater samples categorized as, SW-Kisumu city: shallow wells located in Kisumu city; SW-Kano plain: shallow wells in Kano plain; BH-Ahero town: boreholes in Ahero town; BH-Public: boreholes in public institutions located in populated neighborhoods; BH-Kano plain: boreholes in Kano plain; and BH-Kano plain 1, 7, 18, 20, and 29: boreholes in Kano plain showing $NO_3^-$ decrease with $\delta^{15}N$-$NO_3^-$ increase, during wet (green symbols) and dry (brown symbols) seasons.

A $NO_3^-$ vs. $Cl^-$ concentration plot is another effective tool to distinguish $NO_3^-$ reduction by a biogeochemical processes from dilution [9,39]. The method is quite useful in cases where two water masses from different groundwater flow paths containing different $NO_3^-$ and $Cl^-$ concentrations mix, and assumes that $Cl^-$ is a conservative element unaffected by the biogeochemical processes occurring in groundwater [39]. A plot of $NO_3^-$ vs. $Cl^-$ is shown in Figure 6 and a theoretical dilution line generated by joining the largely unpolluted BH-Kano plain to most polluted SW-Kisumu city as end members. Any groundwater resulting from the mixing of the two end members should lie closely to this line. On the other hand, groundwater affected by $NO_3^-$ removal via denitrification should appear below the theoretical dilution line due to $NO_3^-$ removal alone [9]. In the current study, samples lying along this line consist of SW-Kisumu city (wet season) and BH-Ahero town (wet and dry seasons), both of which were on the high $NO_3^-$ and $Cl^-$ concentration range. Along this line

were also the boreholes in low $NO_3^-$ and $Cl^-$ concentration range, which were located in the recharge (Nandi hills) areas and the sparsely populated parts of Kano. The samples plotting along the $Cl^-$ axis (Figure 6), which include the Awasi boreholes with high $Cl^-$ but significantly low $NO_3^-$ may mainly be portraying a salinization effect. At the same time several samples lie below the theoretical mixing line showing a significantly lower $NO_3^-$ concentration than would be expected if dilution was the only controlling factor for $NO_3^-$ concentration. Samples clearly indicating the $NO_3^-$ reduction process of denitrification include all of SW-Kisumu city (dry season), BH-Public (BH: 11, 26, and 32), and one of the BH-Kano plain (BH: 31). Together with the low $NO_3^-$ concentration, the samples recorded highly enriched $\delta^{15}N$ values (Figure 5), thus corroborating denitrification as a key process responsible for the $NO_3^-$ reduction.

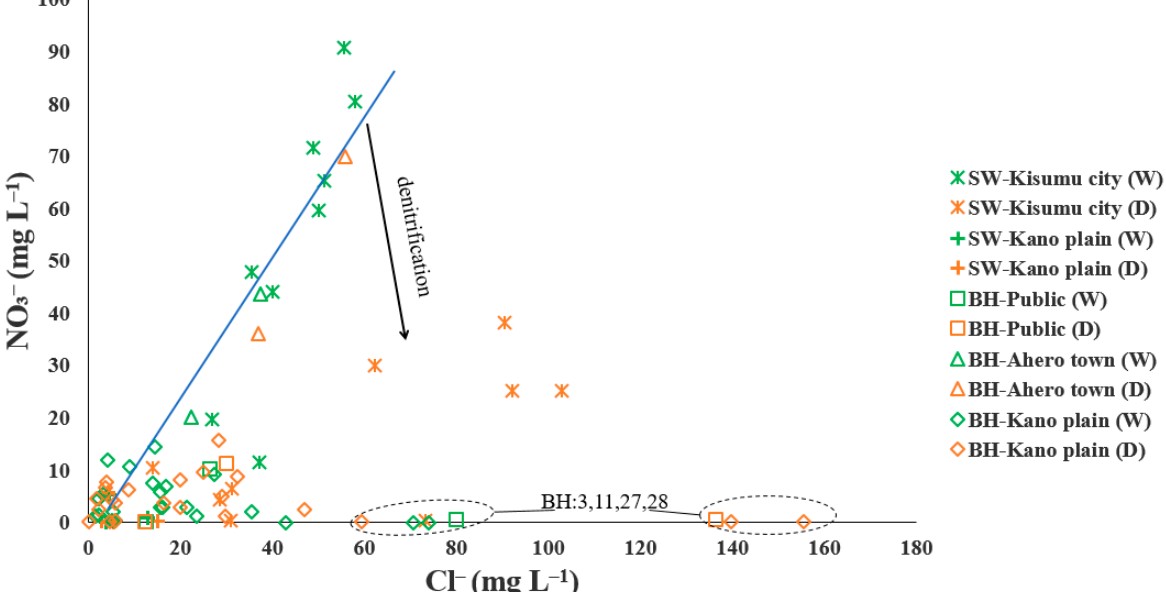

**Figure 6.** $Cl^-$ vs. $NO_3^-$ concentrations of groundwater samples. SW-Kisumu city: shallow wells located in Kisumu city; SW-Kano plain: shallow wells in Kano plain; BH-Ahero town: boreholes in Ahero town; BH-Public: boreholes in public institutions located in populated neighborhoods; and BH-Kano plain: boreholes in Kano plain, during wet (green symbols) and dry (brown symbols) seasons, for determination of $NO_3^-$ removal mechanisms. Blue line is the dilution line, while the black arrow indicates samples undergoing $NO_3^-$ removal through denitrification.

Based on Figure 4, Figure 5, and Figure 6, it could be concluded that denitrification was responsible for $NO_3^-$ attenuation observed in all the SW-Kisumu city (dry season), BH-Public (both seasons), and several of the BH-Kano plain (BH: 1, 7, 18, 20, 29, and 31) samples. It is noteworthy that the data indicates denitrification occurring in oxic conditions (DO ≥ 1.6 in dry season), however, it has been demonstrated that groundwater denitrifiers can be active in anoxic microsites while the bulk of groundwater is well oxygenated [40]. In the case of BH-Ahero town, an increasing $NO_3^-$ concentration corresponding to $\delta^{15}N$ increase was observed moving from wet to dry season (Figure 5). This may mean that an enriched $\delta^{15}N$ source is responsible for the $\delta^{15}N$ enrichment as opposed to denitrification, which in this case should be the urban sewage source. Furthermore, Figure 6 corroborates that denitrification may not be a major process in the BH-Ahero town samples (and confirms sewage source) because they plotted along the dilution line on the high $NO_3^-$ and $Cl^-$ concentration range in both seasons.

The significantly higher $NO_2^-$ content, corresponding to lower $NO_3^-$ content observed in the dry season indicates that denitrification may not be the only N conversion process taking place in groundwater in the study area. Partial nitrification occurs in waste water contaminated systems under high $NH_4^+$, high temperature, and limited oxygen conditions, which favors ammonium oxidizing

bacteria but inhibits the nitrite oxidizing bacteria, resulting in accumulation of $NO_2^-$ but reduced $NO_3^-$ production in the system [41,42]. The fact that most groundwater samples in this study plot in the manure and sewage source domain, coupled with the sanitation problem in the study area implied that sewage (urban and domestic) contamination of groundwater was common. This should result in elevated $NH_4^+$ concentrations. However, in this study $NH_4^+$ concentrations were generally low while $NO_2^-$ concentrations were elevated (Table 1), indicating that (partial) nitrification was occurring. Although DO levels reported in this study (mean: 4.1 mg $O_2$ $L^{-1}$) were somewhat above the 1 mg $O_2$ $L^{-1}$ threshold reported by Wyffels et al. [41], for initiating partial nitrification to sustain $NO_2^-$ accumulation, these were snapshot measurements and might therefore vary with time, location, and depth. In addition, the relatively higher temperatures (mean: 26 °C (SW) and 28 °C (BH)) observed in this study is another favorable condition for the high rate of $NO_2^-$ production witnessed [41,43]. For instance, Pynaert et al. [43] reported a temperature of 26 ± 1 °C as favorable for the high activity of $NH_4^+$ oxidizing bacteria, resulting into high production of $NO_2^-$. Hence partial nitrification of sewage derived $NH_4^+$ may thus be another explanation for low $NO_3^-$ and the significantly higher $NO_2^-$ observed in the dry season. However, further studies are required to establish the inorganic nitrogen dynamics (e.g., nitrification, denitrification, and anammox) in groundwater in the area.

To discriminate manure from sewage sources and at the same time overcome any bias in $NO_3^-$ source apportionment, which might have been occasioned by denitrification, boron (B) isotopic values were determined for the three potential $NO_3^-$ sources and for representative groundwater samples. B and $\delta^{11}B$ values were analyzed in a selection of representative groundwater samples from: Kisumu shallow wells located in the informal settlements (SWa); Kisumu shallow wells in newly planned estates (SWb); boreholes in public institutions located in populated neighborhoods (BH-Public); boreholes located in Kano plain (BH-Kano plain); and boreholes situated in the Ahero town (BH-Ahero town).

The boron concentration was highest in the inorganic fertilizers (15-2500 µg $L^{-1}$) followed by manure (127-581 µg $L^{-1}$), sewage (25-46 µg $L^{-1}$), and groundwater (16-34 µg $L^{-1}$). The $\delta^{11}B$ values of the three sources were fertilizers ranging from −4.3‰ to +7.8‰; sewage from +16‰ to +22‰; and in manure from +31‰ to +37‰. The fertilizer and manure $\delta^{11}B$ values fall in the literature range of −9‰ to +15‰ reported by Widory et al. [11] and Komor et al. [44] in fertilizers and +15.3‰ to +42.1‰ reported by Widory et al. [45] in manure. The sewage $\delta^{11}B$ values obtained in this study were higher than values reported in previous studies [12] ranging from −3.5‰ to +13‰. However, $\delta^{11}B$ of our sources showed clear differences, which allowed contrasts to be made between the three sources.

A plot of $\delta^{11}B$ versus 1/B (Figure 7) show the groundwater samples plot close to the sewage and manure source boxes. The SWa show a strong $\delta^{11}B$ signature of a sewage source while BH-Public and BH-Ahero town also aligned themselves to the sewage source. The BH-Kano plain and SWb showed a $\delta^{11}B$ signature similar to the manure sources. SWa were located in the densely populated informal settlements (Obunga, Nyalenda, and Manyatta) of Kisumu city, which lacks formal sanitation systems but are characterized by the use of pit latrines, open defecation, and landfills [7]. In addition, sewer contaminated surface water canals are common whose effluents together with the landfill and pit latrine wastes can easily leach $NO_3^-$ into the city's shallow groundwater. BH-Public are situated in public institutions in populated neighborhoods while BH-Ahero town is located along the Kisumu-Nairobi highway. These two locations also have high pit latrine density (in every homestead and institution) while open surface channels drain the Ahero town effluents due to lack of conventional sewer system. The lack of proper sanitation systems is the reason why sewage dominates groundwater $NO_3^-$ input in the three locations. This agrees with the highly enriched $\delta^{15}N$-$NO_3^-$ values obtained in SWa, BH-Public, and BH-Ahero town. SWb on the other hand are located in newly planned estates in Kisumu (Migotsi and Kibos) with high-rise apartments, which are connected to the city sewer system. However, these estates neighbor the peri-urban zone of Kibos where small scale mixed farming and free-range livestock keeping are common. BH-Kano plain are situated in the rural parts of the Kano plain characterized by small scale mixed farming of food crops and livestock. Therefore, in the two locations (SWb and BH-Kano plain), animal manure use in farming and free-range livestock

keeping, which leaves animal wastes littering streams and the land surface can easily leach $NO_3^-$ into groundwater aquifers. The $\delta^{11}B$ data successfully disentangles the manure and sewage sources and augments hydrochemistry and $NO_3^-$ isotope findings in identifying the sources of groundwater $NO_3^-$ contamination in Kisumu city and its surroundings.

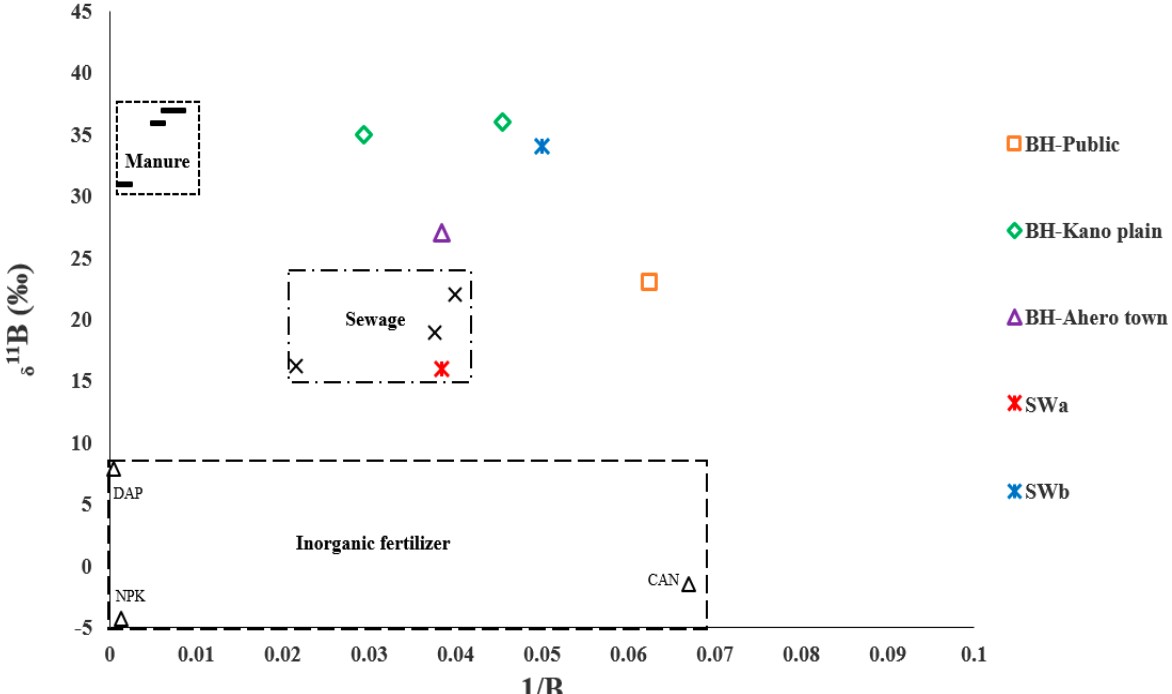

**Figure 7.** $\delta^{11}B$ versus 1/B concentration in selected borehole (BH) and shallow well (SW) samples. Value ranges of local sources of manure, sewage, and inorganic fertilizer are shown in boxes. BH-Public: boreholes in public institutions located in populated neighborhoods (brown squares); BH-Kano plain: boreholes located in the rural Kano (green diamonds); BH-Ahero town: boreholes situated in Ahero town (purple triangles); SWa: Kisumu shallow wells located in informal settlements (red stars); and SWb: Kisumu shallow wells located in newly planned estates (blue stars).

## 4. Conclusions and Recommendations

A triple isotope approach indicated that $NO_3^-$ contamination of groundwater in Kisumu city and its surrounding is largely driven by inadequate sewage infrastructure and animal manure application from either farming or free-range livestock keeping in the rural areas. However, in situ $NO_3^-$ attenuation by denitrification and/or dilution concurrently helps to minimize the $NO_3^-$ loading. On the other hand partial nitrification is likely responsible for accumulation of $NO_2^-$ in the groundwater system. It is however, necessary to develop more process-based research for an in-depth understanding of groundwater N fate in the area. Expansion and improvement of waste-water sanitation should urgently be implemented in the region to avoid further deterioration of groundwater sources.

**Supplementary Materials:** The following are available online at http://www.mdpi.com/2073-4441/12/2/401/s1, Table S1: Hydro-chemical and isotopic parameters for shallow wells (SW) and boreholes (BH) in Kisumu and the Kano plains during the wet season (May–July, 2017); values <0.04 for $NO_2^-$ and $NO_3^-$, and <0.01 for NH4+ indicate attributes below detection limit; "−" represents samples not analyzed., Table S2: Hydro-chemical and isotopic parameters for shallow wells (SW) and boreholes (BH) in Kisumu and the Kano plains during the dry season (February, 2018); values <0.04 for $NO_2^-$ and $NO_3^-$, and <0.01 for NH4+ indicate attributes below detection limit; "-" represents samples not analyzed.

**Author Contributions:** Conceptualization, B.N., P.B. and S.M.; Data curation, B.N. and P.B.; Formal analysis, B.N. and P.B.; Funding acquisition, B.N., P.B. and S.M.; Investigation, B.N. and S.M.; Methodology, B.N., P.B. and S.M.; Project administration, P.B. and S.M.; Resources, P.B. and S.M.; Software, B.N.; Supervision, P.B. and S.M.;

Validation, P.B. and S.M.; Visualization, B.N. and P.B.; Writing—original draft, B.N.; Writing—review & editing, B.N., P.B. and S.M. All authors have read and agreed to the published version of the manuscript.

**Funding:** This research was funded by VLIRUOS (Belgium) TEAM project: "Improved management for nitrate pollution in the Lake Victoria catchment of Kenya-*ZEIN2016PR423*".

**Acknowledgments:** We acknowledge the human resources and field facilitation support given by the Ministry of Water and Irrigation (Headquarters-Nairobi) and Water Resources Authority (Kisumu).

**Conflicts of Interest:** The authors declare no conflicts of interests.

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
