# Peer review of "Tracking Sources and Fate of Groundwater Nitrate in Kisumu City and Kano Plains, Kenya"

_water, doi:10.3390/w12020401_

Round 1

Reviewer 1 Report

The manuscript is very well designed, written and presented. The subject is interessting and dessrved to be published in this journal as it is. 

Author Response

We take note of the observation by this reviewer that English language and style are fine and require minor spell checks. The reviewer's overall conclusion is that the manuscript is very well designed, written, presented and deserves to be published as it is. This is well appreciated.

Reviewer 2 Report

The comments are attached in file.

Reviewer 3 Report

Revision of the manuscript “Tracking sources and fate of groundwater nitrate in Kisumu, Kenya and surroundings”

The manuscript deals with an actual problem of groundwater contamination with nitrate. It attempts to identify the sources of groundwater contamination in urban and rural areas in Kenya. Overall, the manuscript describes an interesting study. I recommend to consider the manuscript for publication after careful revision. Please find my specific comments below.

The first problem appears with the English since it needs a style revision/correction.

Sentences are very long, confusing and hard to read. You over use attributive nouns. For example:

“In addition, the supply of Lake Victoria water is limited because of the low production capacity of the city’s water service provider grappling with increased treatment costs caused by pollution and eutrophication of Lake Victoria.”

Or the following sentence:

“This has made the piped lake water unaffordable to the majority of the residents in the city and its surrounding areas, leading to an increased reliance on low cost (hand dug) shallow wells or mostly donor funded community water supply boreholes.”

I also suggest to split your text into paragraphs. It will facilitate reading and understanding. For example in subsection 2.2: L151 (The d15N- and.....) / L176 (The water analysis...).

I suggest to use terms like nitrate, nitrogen, etc, instead of NO3-, N in the text.

Editorial errors must be corrected (e.g. L95: km2, HNO3- instead of HNO3-, ion notation in Table 1, double spaces, square brackets for citations (e.g. L269), etc.).

Title: The title needs to be revised. I suggest: “Kisumu city and Kano plains, Kenya”, instead of “Kisumu, Kenya and surroundings”.

L13: groundwater point – this term is wrong. Do you mean groundwater wells?

L15: No comma is used between the month and the year when they are the only two elements in the date. From https://www.thepunctuationguide.com/comma.html

L41: Be more specific. Do you mean here private, small capacity wells? Groundwater is not a basic, not an alternative source of drinking water.

L48: fuelled -> driven?

L65: potential threat to groundwater nitrogen contamination – rewrite, e.g. “potential threat to groundwater quality” or “potential source of...”

L95: longitude 34°35' – 34°55'E (not ;)

L95: 35.5% of the city is covered by the lake? Something is wrong with this sentence

L102-119: needs restructuring: describe geology first, than hydrogeology, groundwater dynamics, next land use

Figure 1b: add the boarders of the study area also here

L162: The equation is attached as a picture of poor quality. It should be tipped in Word Equation Editor or similar.

L223-224: organic surface water contaminants – what do you mean?

Table1: Statistics methods should be described in the methods. ANOVA is not mentioned in the text, only in the table. In L208 you suddenly write about p-values, which does not make sense, if you donk link them with a specific statistical analysis.

L251 and further: Piper diagram, not piper diagram

L258: clear mineralisation ???

L269: where does the saline water comes from?

Figure 2: It would be helpful to present you sampling points in similar manner, so one can easily see, where the results comes from.

in %mval, not %

L289: It is clear that – rewrite, for example: “As expected, ...”

L295-298 and 310-311 – repeated

L313: Borehole NO3-, borehole concentration – use other terms, these are just wrong

References: Why don’t you refer to your own paper? http://www.ini2016.com/pdf-papers/INI2016_Nyilitya_Benjamin.pdf

Supplementary material: How did you deal with the values below detection limit. There are many of them and in your summary statistic you give numeric values for minimums.

Reviewer 4 Report

Nyilitya et al. conducted a study on tracking sources and the fate of groundwater nitrate in Kisumu, Kenya. They collected some valuable data. However, the manuscript does not offer any significant contribution to science.  There are a lot of speculative statements. Authors need to strengthen their presentation and decision to improve the manuscript. For these reasons, I am unable to recommend this manuscript for publication.

Specific comments are as follows.

 Line 46: Neural tube defects are a severe threat due to nitrate in drinking water. The authors should mention this in the manuscript. Authors should also include here that nitrogen persists in groundwater, so it becomes a severe threat to water quality.

Brender, J.D.; Weyer, P.J.; Romitti, P.A.; Mohanty, B.P.; Shinde, M.U.; Vuong, A.M.; Sharkey, J.R.; Dwivedi, D.; Horel, S.A.; Kantamneni, J.; et al. Prenatal nitrate intake from drinking water and selected birth defects in offspring of participants in the National Birth Defects Prevention Study.  Health Perspect. 2013, 121, 1083–1089.

Hot spots and persistence of nitrate in aquifers across scales, D Dwivedi, B Mohanty - Entropy, 2016

2. Lines 61-65: Authors talked about the conventional sanitation system. How does contamination happen to the groundwater system? What is the role of the vadose zone? In the literature, it has been demonstrated that linked surface-water-groundwater and soil water can lead to groundwater contamination. The vadose zone can remove microbial or nitrogen contamination. For example, see the following article:

Impact of the linked surface water-soil water-groundwater system on transport of E. coli in the subsurface, D Dwivedi, BP Mohanty, BJ Lesikar - Water, Air, & Soil Pollution, 2016

3. Lines 230-231: Biogeochemical processes and redox states of the aquifers are missing. Authors have talked about microbial pathways but have not included explicitly. Also, it is not clear how DO and nitrogen species are linked in the aquifer. There are several papers in the literature that have demonstrated such relationships. For example, see the following articles and enhance your discussion accordingly. Hot spots and hot moments of nitrogen in a riparian corridor D Dwivedi, B Arora, CI Steefel, B Dafflon, R Versteeg - Water Resources Research, 2018 Impact of intra-meander hyporheic flow on nitrogen cycling, D Dwivedi, IC Steefel, B Arora, G Bisht - Procedia Earth and Planetary Science, 2017 Geochemical exports to river from the intrameander hyporheic zone under transient hydrologic conditions: East River Mountainous Watershed, Colorado, D Dwivedi, CI Steefel, B Arora, M Newcomer… - Water Resources Research, 2018 Influence of hydrological, biogeochemical and temperature transients on subsurface carbon fluxes in a flood plain environment, B Arora, NF Spycher, CI Steefel, S Molins, M Bill… - Biogeochemistry, 2016

4. Lines 102-103: Show groundwater flow direction on the associated Figure.

5. Table 1 is hard to follow. Please make figures (or heatmaps) to show these statistical measures

6. Lines 297-298: Because of the limited sewerage infrastructure network in Kisumu, untreated sewage discharges into urban streams is common. How do you know what percentage is being removed due to natural processes (see Comment 2)?

7. Line 313: “may also be contributing to the observed NO3- levels”? How do the authors know?

8. Lines 389-390: Based on Figures 4, 5, and 6, it can be concluded that denitrification was responsible for NO3- attenuation? It is not clear from these figures. See papers gives in Comment 3. These papers talk about how nitrate is removed from the system. Authors should include in their discussion nitrate removal mechanisms.

Author Response

Please see the attachment, a letter from the editorial office concerning this reviewer

Round 2

Reviewer 2 Report

Following the comments, the paper was revised appropriate.

Author Response

The reviewer's comment that "the paper was revised appropriately", is well-noted and appreciated. 

Note that, the paper underwent English grammar editing by a native English speaking colleague.

Reviewer 3 Report

After the first revision and corrections the manuscript have been improved.

Nevertheless the manuscript still needs further revisions as listed below:

Figure 1: The figure caption (b) is hidden behind the figure.

L143-146: Please rewrite this two sentences, because they are repetition of the same information. “The aim was to identify production BHs… They included BHs…”. Or substitute “they” with “the final list of observation points”.

L150-152: “Water samples were taken from production BHs and SWs. In case the well wasn’t pumping prior to the sampling exercise, the well was purged to ensure the representativeness of a sample.”

L152-153: 11 μm whatmann filters -> should be “11 µm filters (Whatman, GE Healthcare Life Sciences, Chicago, IL, USA)” or similar.

Table1: WHO stands –> WHO limit? world health organisation –> World Health Organisation

L273: Do you mean “groundwater recharge”?

L275: Hydrochemical facies should be corrected: eg. Ca+Mg-HCO3 is either Ca-Mg-HCO3 or (more precisely) Ca2+-Mg2+-HCO3-

L276: I insist to refer to your previous paper. Here would be an appropriate place, as in your previous study you describe chemistry of Nyado River.

L287 and further: The formula of sulphate must be corrected to SO42-

L343: The term “depths above 150m” is unusual. I suggest “depths greater than 150 m”.

Figure 2: The Piper diagram is based on parameter concentrations expressed in equivalents per volume. Therefore, the common way is to present the units as “% meq/l” (percent of total equivalents per liter) not “%”. https://pubs.usgs.gov/wsp/wsp2254/pdf/section5.pdf (Page 179)

Figure 7: Again, a lot of attributes make your sentences hard to read. Why not: “δ11B versus 1/B concentration in selected points. Range of the values of local sources of manure, sewage and inorganic fertilizer is shown in boxes”?

L510: Either “animal manure application from either farming or free-range livestock keeping in the rural areas” or “either animal manure use from farming or free-range livestock keeping in the rural areas”.
